# Ultrastiff metamaterials generated through a multilayer strategy and topology optimization

Yang Liu[1,2], Yongzhen Wang[1], Hongyuan Ren [1], Zhiqiang Meng[2], Xueqian Chen[1], Zuyu Li [3,4] ✉, Liwei Wang [5], Wei Chen [5], Yifan Wang [2] & Jianbin Du [1] ✉

Metamaterials composed of different geometrical primitives have different properties. Corresponding to the fundamental geometrical forms of line, plane, and surface, beam-, plate-, and shell-based lattice metamaterials enjoy many advantages in many aspects, respectively. To fully exploit the advantages of each structural archetype, we propose a multilayer strategy and topology optimization technique to design lattice metamaterial in this study. Under the frame of the multilayer strategy, the design space is enlarged and diversified, and the design freedom is increased. Topology optimization is applied to explore better designs in the larger and diverse design space. Beam-plate-shell-combined metamaterials automatically emerge from the optimization to achieve ultrahigh stiffness. Benefiting from high stiffness, energy absorption performances of optimized results also demonstrate substantial improvements under large geometrical deformation. The multilayer strategy and topology optimization can also bring a series of tunable dimensions for lattice design, which helps achieve desired mechanical properties, such as isotropic elasticity and functionally grading material property, and superior performances in acoustic tuning, electrostatic shielding, and fluid field tuning. We envision that a broad array of synthetic and composite metamaterials with unprecedented performance can be designed with the multilayer strategy and topology optimization.

Lattice metamaterials show many advantages in the design of synthetic and composite metamaterials due to sophisticated topologies and length scales. Nowadays, lattices have been applied to many aspects of practical engineering, such as mechanical, acoustic, electromagnetic, and optical fields, and so on[1–4].

According to the dominant structural elements, lattice metamaterials can mainly be divided into three categories, that is, beam-, plate-, and shell-based lattices, corresponding to line, plane, and surface in structural geometry, respectively. Different types of lattices have different advantages as well as disadvantages. The beam-based lattices have been widely studied in theoretical research, numerical simulation, and experiment[5–9], and particular functional properties, such as isotropy[10–12], auxeticity[13], and chirality[14] can be achieved thanks to the flexibility of the line geometrical form. However, beam-based lattices are prone to stress concentration at the point joints between the strut elements where flaws or imperfections are more likely to occur. The plate-based lattices show superiority in mechanical performance such as stiffness and strength. Lattices with combinations of cubic and octet

[1]School of Aerospace Engineering, Tsinghua University, Beijing, PR China. [2]School of Mechanical & Aerospace Engineering, Nanyang Technological University, Singapore, Singapore. [3]School of Automation, Guangdong University of Petrochemical Technology, Maoming, China. [4]School of Mechanical and Mechatronic Engineering, University of Technology Sydney, Ultimo, New South Wales, Australia. [5]Department of Mechanical Engineering, Northwestern University, Evanston, Illinois, USA. ✉e-mail: lizuyu@gdupt.edu.cn; dujb@tsinghua.edu.cn

plate geometries can reach the Hashin-Shtrikman upper bound[15] for the isotropic elastic moduli, including Young's modulus, bulk modulus, and shear modulus[16–19]. The experimental results showed that the stiffness and compressive strength of the plate lattice are always higher than those of the beam-based lattices with the same mass[17]. Elastic isotropy can also be realized in plate-based lattices by combining certain lattice topologies, such as simple cubic, body-centered cubic, and face-centered cubic[17]. Besides, plate-based lattices as metamaterials also demonstrated excellent sound and mechanical energy absorption performance[20]. Nevertheless, imperfections can also be induced around connections and corners of plates, where risks of stress concentration and bulking potentially lie. Moreover, the tunable dimension for the plate element forming plate-based lattices is generally limited to the thickness due to the geometrical form of the plane. The lack of tunability restricts the applicability of lattice metamaterials for complex engineering scenarios. The shell-based lattices, whose cells are composed of continuous and smooth-curved surfaces, demonstrated high strength, and stiffness at low density[21]. Compared with plate-based lattices, shell-based lattices enjoy continuity and smoothness of the geometrical form of the surface. Theoretical and numerical analyses showed that the continuity and smoothness of the surface are very important in suppressing local buckling[22]. The triply periodic minimal surfaces (TPMS) have attracted widespread attention in the materials and engineering fields due to their neoteric, symmetrical structures, and excellent mechanical properties[23–25]. TPMS naturally inherits the advantages of the surface. The mean curvature at each point of the TPMS is zero, which is a continuous and smooth surface and has no sharp edges[26–28]. The smooth transition between different crystal cells of the TPMS lattice will reduce the occurrence of

stress concentration, thereby improving the overall mechanical properties[29–31]. Studies showed that the elasticity of materials with particular topologies of TPMS can approach the Hashin-Shtrikman upper limit[32,33]. The advantageous geometrical form of the smooth surface of TPMS also attributed to superior performance in large geometrical deformation, leading to better energy absorption capacity[34–39]. Despite a series of excellent properties of shell-based lattices, their topological configurations usually have fewer variations, and tunable dimensions are limited to the shape[40] and thickness[41].

To enrich the diversity and explore the better performance of lattice metamaterial, we propose a multilayer strategy to enlarge and diversify the design space and associated topology optimization to effectively explore the vast design space for target-optimized performance. Specifically, the multilayer strategy can be realized through scaling, transforming, or hybridizing a series of single-layer elements to form nested models (Fig. 1a). While the multilayer strategy leads to higher design freedom, it also imposes challenges on the design process to identify promising structures in a large design space. To address this, we further develop a topology optimization technique to optimize topological configuration and improve the structural performance of lattices. Topology optimization targets on redistributing material to achieve reasonable configuration with prescribed objectives and constraints[42,43]. To date, topology optimization has been applied to metamaterial design in many aspects[44–50]. As can be seen in Fig. 1b, optimized solutions in this study automatically converge into a comprehensive combination of beam, plate, and shell, which can fully utilize material with different structural archetypes. For example, the implementation scheme of the multilayer construction and topology optimization is illustrated in Fig. 1c. A unit cell of Schwarz P forming the

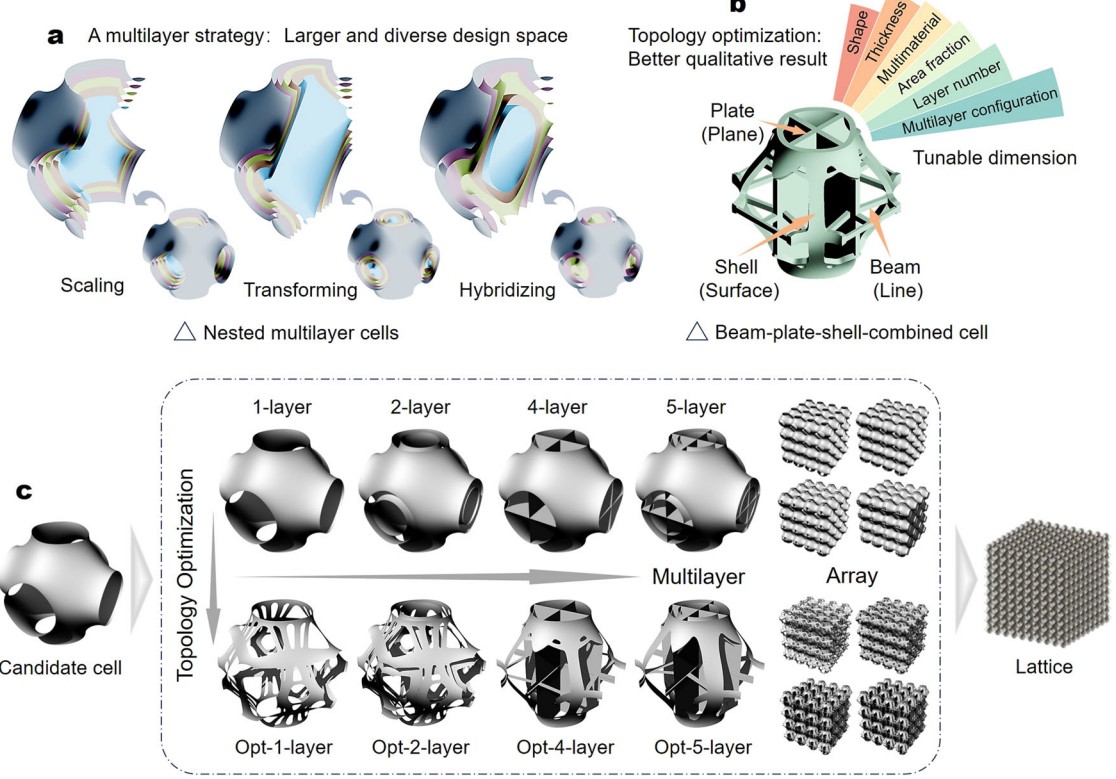

**Fig. 1 | Schematic diagram of a multilayer strategy and topology optimization for the design of lattice metamaterial. a** Illustration for the multilayer strategy. The nested multilayer cells can be constructed by scaling, transforming, or hybridizing a series of plane or surface layers (represented in different colors). **b** illustration for topology optimization. Through topology optimization, the multilayer model automatically converges into a beam-plate-shell-combined

complex with reasonable material distribution. **c** illustration for the technological route of the proposed design method. A candidate cell is first designated, and then the candidate cell is subjected to the design process. The horizontal arrow indicates the multilayer construction and the vertical arrow indicates the topology optimization process. After the design procedure, the lattice can be arrayed.

lattice is first identified as the candidate. Through scaling, transforming, and hybridizing, a nested multilayer model can be developed from the candidate cell. With prescribed objectives and constraints, the candidate cell and its multilayer variants are optimized. Finally, new lattices can be assembled with given unit cells. Through a series of analytical analyses, numerical simulations, as well as physical experiments, our findings show that the mechanical performances of optimized lattices demonstrate considerable improvement. On the other hand, the introduction of the multilayer strategy and topology optimization also brings a series of tunable dimensions, including shape, thickness, multimaterial, layer number, multilayer configuration, and area fraction (the ratio of the area of solid region and the area of the whole surface). With excellent tunability, particular structural mechanical properties, such as isotropic elasticity, and functionally grading stiffness, can be achieved conveniently for lattice metamaterial. We also envision that the design method in this study will bring new opportunities in the application of acoustic absorption design, electrode design, fluid channel design, and so on.

## Results

The multilayer strategy is inspired to expand and diversify the design space as the design freedom for a single-layer model is limited. With a tailored multilayer configuration, the multilayer model is possible to achieve better results compared with a single-layer model at the same density. The topology optimization is targeted to redistribute material more efficiently with certain objectives, specifically, maximizing the total stiffness in this study.

### Single-layer design

We first studied the design of single-layer lattices. Three types of single-layer TPMS models are discussed, that is, Schwarz P, IWP, and Neovius. We compared the mechanical performances of the original designs and optimized results at the same relative densities to ensure fairness. As can be seen from the comparisons of normalized Young's moduli and yield strengths in Fig. 2c, d, respectively (original data in Supplementary Fig. 4 and Supplementary Table 1 in the supplementary information (SI) document), the "Opt-" results basically demonstrate improvements in stiffness and strength. The optimized geometries generally see uniaxial mechanical improvements as the optimizer concentrates more material in resisting loads from one unique direction (see strain energy distribution and uniaxial deformation comparisons in Supplementary Fig. 5). In particular, the Opt-Neovius model achieves remarkable stiffness and strength increases (by around 50%) at the same relative density, showing capability in reaching the theoretical Voigt bound (Fig. 2a). However, digging holes on the surface with topology optimization does not necessarily guarantee better performance. Several factors, such as the thickness, deformation mode, stress concentration and local deformation, may result in structural deterioration (detailed discussion in Supplementary Note 5 in SI). Despite improvements through material redistribution, topology optimization is restricted to performing on the single surface. The optimization profit is destined to be limited due to limited design freedom.

### Multilayer design

Compared with the single-layer design, the use of multiple layers greatly enlarges the design space and enables more diverse geometries. Together with the powerful topology optimization technique, better results can be efficiently found in a vast design space. We elaborated on the implementation and advantages of the multilayer strategy and topology optimization, using the Schwarz P candidate as an example. Note that this scheme is also applicable to any other plate- and surface-based models. We defined a P set, including the P-1 (original Schwarz P), -2, -4, and -5, as well as their optimized results under uniaxial loading, i.e., the Opt-P set, including the Opt-P-1, -2, -4, and -5

(see their performances in the material property space in Fig. 2b). All the optimized results were obtained with the same area fraction and tuned into the same mass through thickness adjustment. Mechanical performances of normalized Young's modulus and yield strength are compared in Fig. 2e, f, respectively (original data in Supplementary Fig. 7 and Supplementary Table 2). As shown, the model P-2 shows slight reductions in both stiffness and strength, while P-4 and P-5 demonstrate considerable enhancements. Similar to the original Schwarz P, P-2 also undergoes bending-dominant mechanical behavior. Since bending mode is much more sensitive to the structural thickness, it is sensible that P-2 shows worse performance with thinner structural thickness at the same relative density. While the thicknesses of P-4 and -5 are smaller than P-2, the tailored multilayer configurations transform the loads resisting mode. Specifically, P-4 and -5 undergo stretching-dominant mechanical behavior. In particular, the vertical plate elements provide sufficient stiffness under unilateral loading. Since the thickness of P-4 is larger than P-5, the vertical plate elements inside P-4 are stronger, as a result, P-4 outperforms P-5 in both stiffness and strength. After optimization, Opt-P-1 and -2 become beam-based though maintaining the original shape outline. Opt-P-4 and -5 turn out to be beam-plate-shell-combined entities after material redistribution, and their stiffnesses are further brought into a higher level (increased by around 50%), particularly Opt-P-4 showing the capability of reaching the Voigt upper bound for anisotropic materials. Generally, the larger the design space, the better the optimized outcome of topology optimization. However, it is noted that Opt-P-5 shows worse static mechanical performances compared with Opt-P-4. This is because the structural thickness of Opt-P-5 is thinner than that of Opt-P-4 with a larger structural area but the same mass. With the same thickness and mass, Opt-P-5 should achieve better or at least the same performance as Opt-P-4 (Supplementary Fig. 8). Opt-P-4 and -5 share similar multilayer construction and optimized beam-plate-shell-combined configurations. It is noted that both the multilayer strategy and topology optimization contribute a lot to achieving ultrahigh mechanical performance. On one hand, the tailored multiple layers develop more load-resisting paths, which helps in finding possible better deformation modes for given loading conditions. On the other hand, topology optimization further rationalizes the material distribution on the multiple layers. In the beam-plate-shell-combined configuration, the vertical plate elements mainly remain as they play an important role in resisting vertical loads. The horizontal plate elements degenerate into beam systems and show axial stretching deformation mode. The beam components pull the shell elements to prevent the curved shell from bending excessively under the compressing loads. In this sense, all three structural primitives are organically connected as a whole and exert their respective advantages efficiently (Supplementary Fig. 9). Therefore, the beam-plate-shell-combined configuration substantiates the contribution of the multilayer strategy and topology optimization to remarkable improvements in mechanical performance.

The above-discussed comparison was based on data from numerical simulations for unit cells. We also studied the scale effect for the discussed lattices. As presented in Supplementary Fig. 10, lattices with three scales are compared, that is, unit cell, 4X4X4 array, and infinite array. As the lattice array becomes denser, the mechanical performance shows improvement, especially for the cases of P-1 and -2 as well as their optimized results. The benefits come from the changeover in loads carrying manner, that is, from bending- to stretching-dominated. As the Gibson-Ashby scaling power-law curve fits show (Supplementary Fig. 10d), the fitting power values for P-1 and -2 are larger than 2, indicating that P-1 and -2 undergo bending-dominated deformation modes. This was alleviated through optimization and the fitting power values decreased from 2.272 and 2.414 to 1.747 (Opt-P-1) and 2.159 (Opt-P-2), respectively. As the lattice array increases, the fitting power values for P-1 and -2 as well as their optimized results further decrease, indicating their deformation modes become stretching-dominated. As for P-4 and

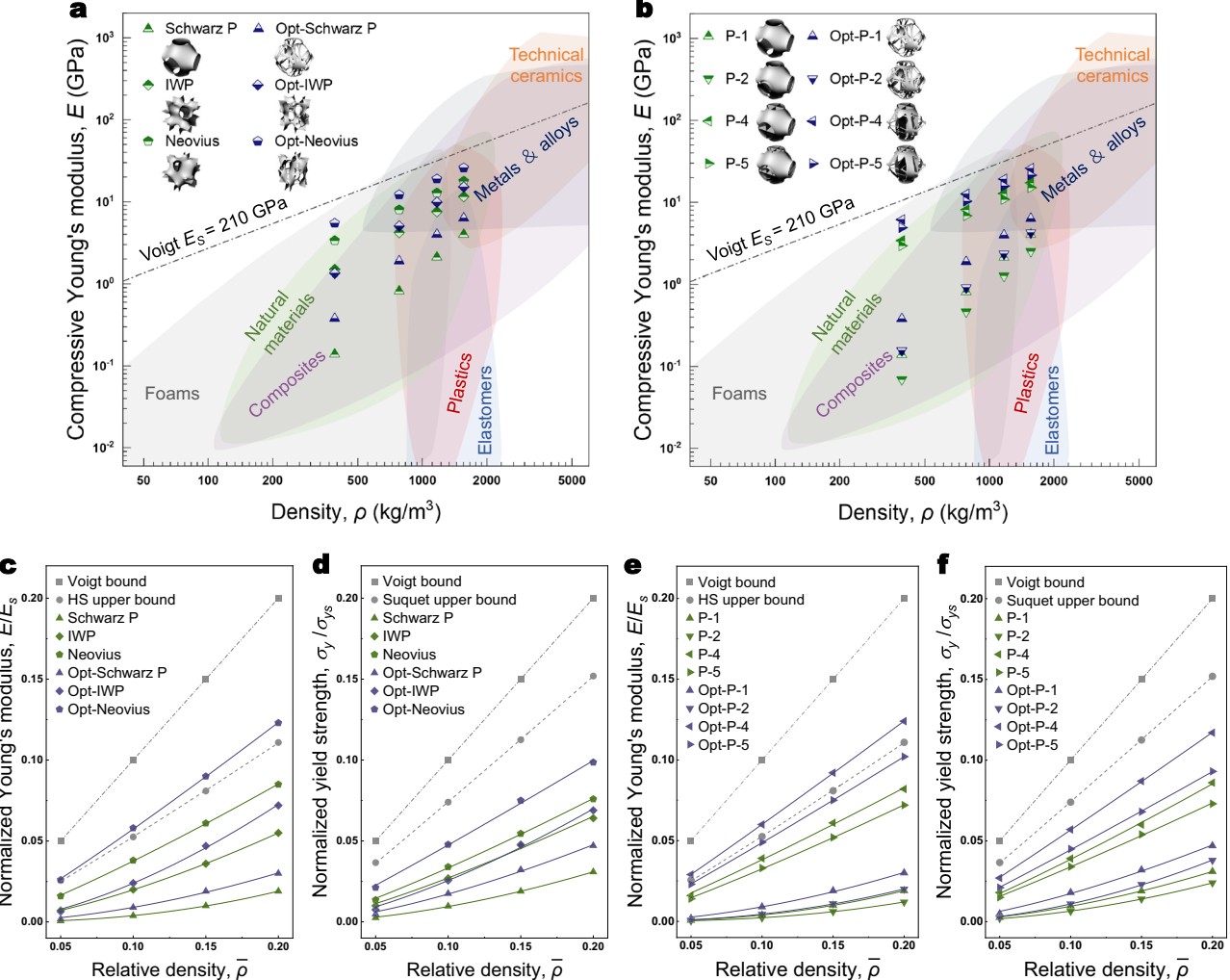

**Fig. 2 | Young's moduli and yield strengths comparisons for various shell-based cells.** The metal iron is assigned as the constitute material, and Young's modulus, yield strength, and density are given as 210 GPa, 400 MPa, and 7800 Kg/m³ respectively. The geometrical cubic length for each cell is 15 × 15 × 15 mm in x-, y-, and z- directions. The Young's modulus and yield strength for the cellular material are calculated by extracting the slope and endpoint of the linear part of the stress-strain curves, respectively. Here, the normalized Young's modulus and yield strength are not the traditional concepts for solid constitute material, but effective concepts for describing mechanical performances for cellular material. The structural thickness was regularized to ensure that the relative density was limited to no more than 20%, otherwise, the thickness might become too large as the density increases thereby violating the presumption of shell. Since topology optimization digs holes on a surface, the mass losses were compensated in thickness for the optimized results by tuning the thickness and area fraction of the surface, leading to the same relative density as the original topologies. Subfigure (**a**) shows the compressive Young's moduli of Schwarz P, IWP, and Neovius and their optimized results in the material property space. Here 'Opt-' refers to optimized results under uniaxial loading. Subfigure (**b**) shows the compressive Young's moduli of the P set and its multilayer variants, and their optimized results in the material property space. Here, the script "n" in "P-" and "Opt-P-" indicates the number of layers. Subfigures (**c**) and (**d**) give the normalized Young's modulus and yield strength vs. the relative density of three different minimal surfaces (i.e., Schwarz P, IWP, Neovius) and their optimized results. Subfigures (**e**) and (**f**) give the normalized Young's modulus and yield strength vs. the relative density of the P set and their optimized results under uniaxial loading. Source data are provided as a Source Data file.

-5 as well as their optimized results, their mechanical performances show independence on the lattice scale, and their corresponding fitting power values are close to 1 for different scales. As such, the shifting of loads carrying manner mainly comes from the multilayer strategy as it creates more load-resisting paths.

The scale effect is the result of boundary conditions. For a unit cell under uniaxial loading, the boundaries without connection to other cells are considered free boundary conditions (FBC). If all cells in a lattice are connected, then the periodic boundary conditions (PBC) are applied. Different boundary conditions may result in different deformation modes and different optimized solutions for the same geometrical model (Supplementary Fig. 11). The models P-1 and -2 are sensitive to the boundary condition. One can see differences in their deformation modes as well as optimized topological configurations

with FBC and PBC (Supplementary Fig. 12). As a result, their mechanical performances rely much on lattice scale. P-4 and -5 are not sensitive to the boundary condition. Therefore, their deformation modes as well as optimized topological configurations are consistent. Generally, the calculated results with PBC are better than those with FBC. It is reasonable as the PBC plays a role of constraint in resisting external loads.

The physical compressing experiments in situ (Supplementary Fig. 13) were conducted to validate the numerical results by subjecting the printed models to uniaxial compression. We studied two scales, that is, unit cell, and 4×4×4 array. For the scale of the unit cell, the P set and their optimized results are printed with four different kinds of material (Fig. 3a), i.e., thermoplastic polyurethane (TPU), nylon (PA12), stainless steel (ss316), and aluminum alloy (AiSi10Mg). Since the yielding strengths for diverse materials can be different and easily

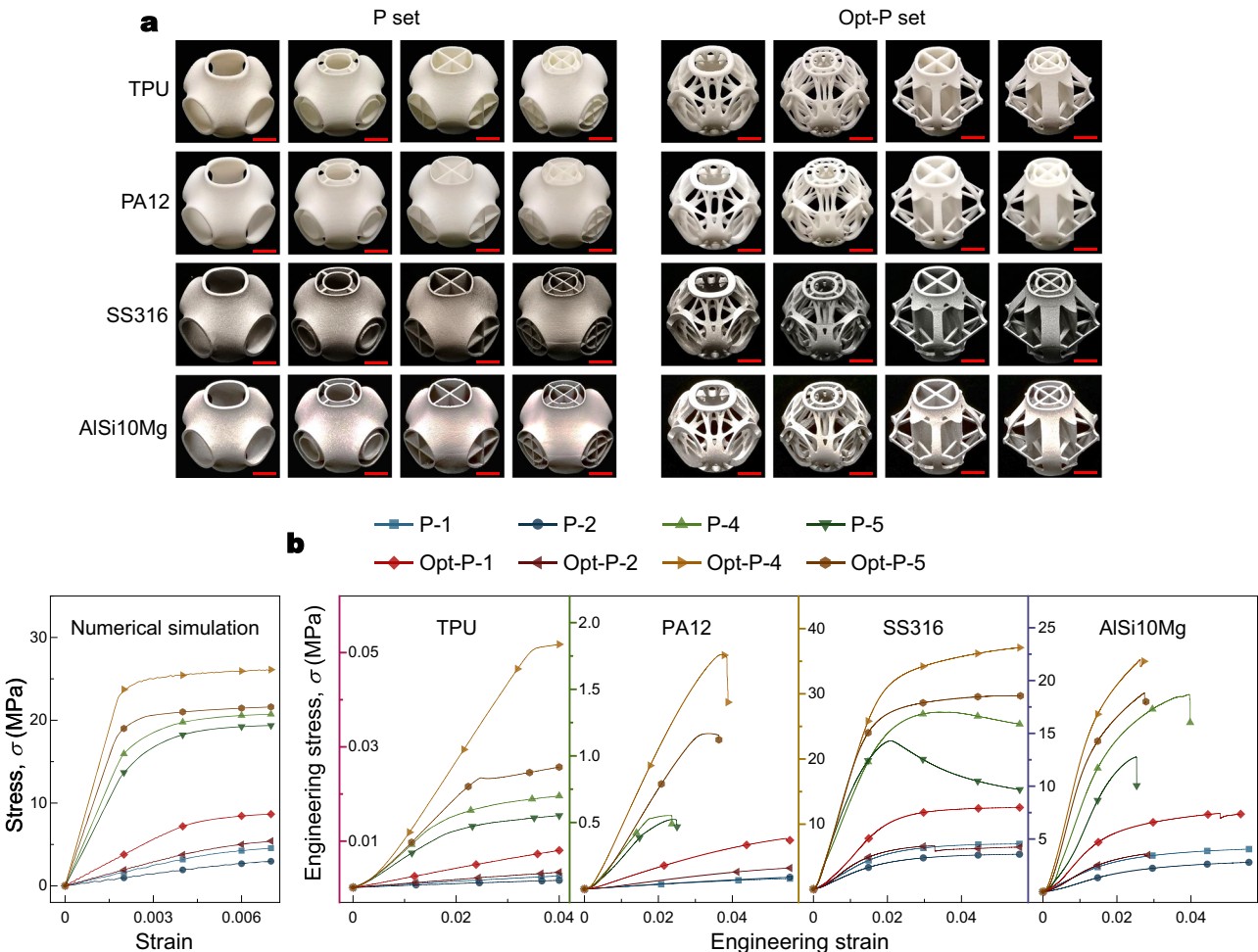

**Fig. 3 | Physical experiment verification for constitutive relationships of unit cells of the P set and their optimized results with four different printing materials. a** Comparison of fabricated models of the P set and Opt-P set with four different types of printing material, that is, thermoplastic polyurethane (TPU), nylon (PA12), stainless steel (SS316), and aluminum alloy (AlSi10Mg). All the red scale bars represent 1 cm. **b** Comparison of numerical simulation results and practical physical experimental results of constitutive relationships of the P set and Opt-P set with four different types of printing material. Source data are provided as a Source Data file.

affected by local printing quality, here we only consider the comparison of uniaxial stiffness in linear elasticity (compressing Young's moduli), i.e., the slopes of all stress-strain curves (original experimental data in Supplementary Fig. 14 and Supplementary Table 3). Thus, the relative proportional ratios of Young's moduli of the P set and Opt-P set should be material-independent. In comparison with the numerical simulations (Fig. 3b), the basic trend of the physical experimental results shows consistency with corresponding numerical results, yet still some minor differences remain for different kinds of printing material. The discrepancies are mainly caused by printing technics and prototyping quality (detailed discussion in Supplementary Note 6 in SI). Also, the anisotropy of additive manufacturing can affect the mechanical performance of the printed models.

For the 4×4×4 array scale, only the experiment for the original Schwarz P and the optimized P set were conducted as P-2, -4, and -5 encountered fabricating difficulties. Particularly, P-4 and -5 are close-cell models, and the printing powder was unable to be removed in postprocess. We considered two array manners, that is, lattices arrayed by unit cells optimized with FBC and PBC, respectively. This is because the stress statuses for cells in different parts of a finite-arrayed lattice are inconsistent (Fig. 4a). The 4×4×4 array P set and their optimized results with two materials, i.e., nylon (PA12), and aluminum alloy (AiSi10Mg), are presented in Fig. 4b. As can be seen from the comparisons in Fig. 4c, d (original data in Supplementary Fig. 15,

Supplementary Table 6, and Supplementary Table 7), the experimental results show consistency with corresponding numerical results, especially for the PA12 results. The relative proportional ratios normalized by the result of the original Schwarz P (100%) are just slightly different from that of the numerical simulations due to the post-printing process of powder cleaning (Supplementary Fig. 17c). The practical stiffness performances for Opt-P-4 and -5 increased by around 3 and 2 times, respectively. The stiffnesses of AiSi10Mg results demonstrate larger gaps with corresponding numerical results, especially the Opt-P-4 and -5 outcomes. This is mainly because of the low printing quality, which leads to coarse surface and large granularity of manufactured models (Supplementary Fig. 18). Consequently, the stiffnesses of printed models are impaired (detailed discussion in Supplementary Note 6 in SI). In brief, the experimental results basically verified the correctness and effectiveness of the numerical simulations, and the physical experimental discrepancies from numerical results can be further narrowed by improving fabricating technology.

## Energy absorption

In some practical engineering situations, many engineering crash scenarios, for instance, require light yet high energy absorption ratio materials under large geometrical deformation. High stiffness can have a positive influence on energy absorption. The impact direction usually is prescribed or predictable in some shock cases like aircraft landings

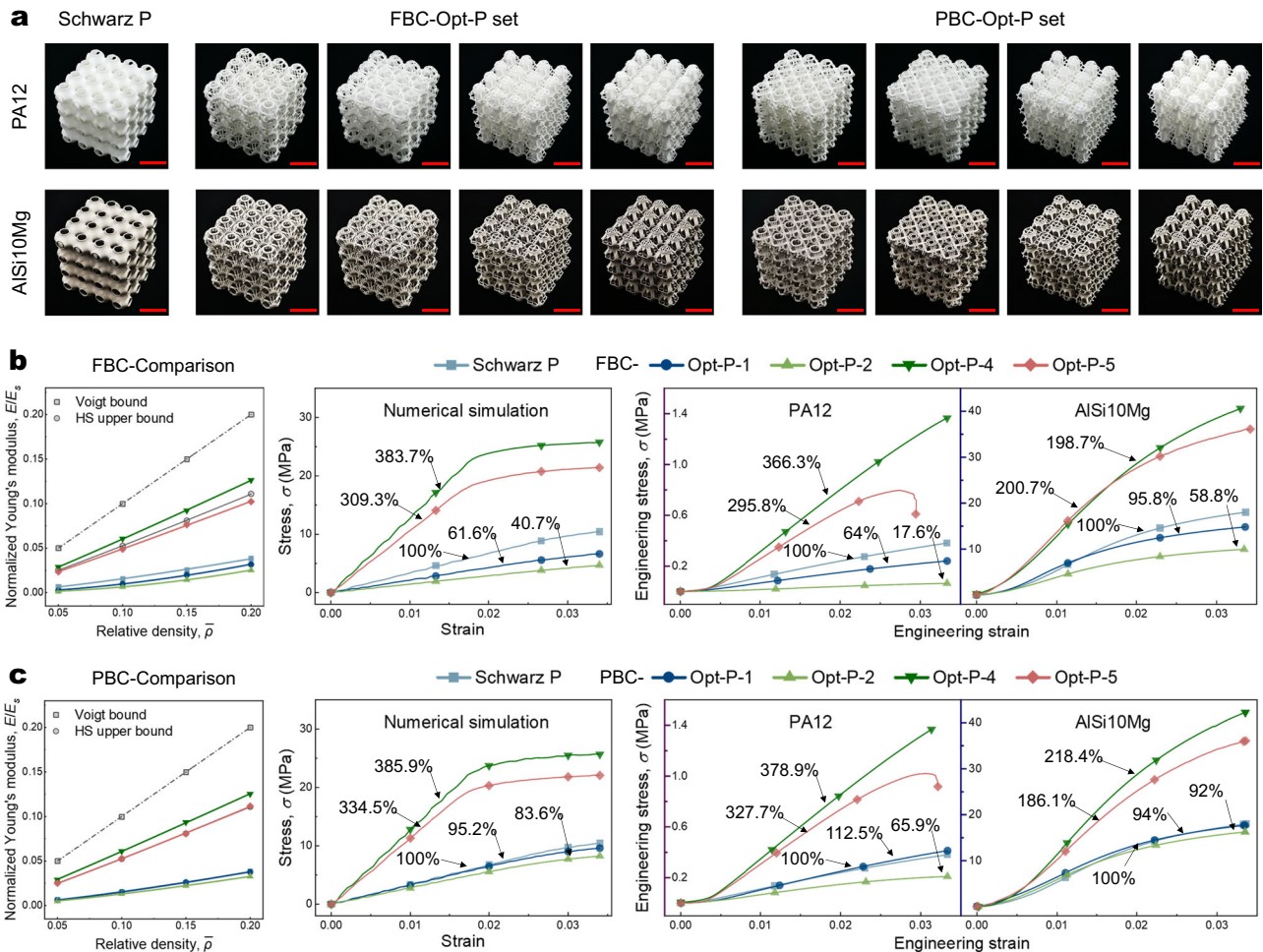

**Fig. 4 | Physical experiment verification for constitutive relationships of 4×4×4 arrays of the P set and their optimized results with two different printing materials. a** Comparison of fabricated models of FBC-Opt-P set and PBC-Opt-P set with two different types of printing material, that is, nylon (PA12) and aluminum alloy (AlSi10Mg). All the red scale bars for the PA12 models represent 6 cm and all the red scale bars for the AlSi10Mg models represent 3 cm. Subfigures (**b**) and (**c**) give comparisons of numerical simulation results and practical physical experimental results of constitutive relationships of the FBC-Opt-P set and PBC-Opt-P set, respectively. Source data are provided as a Source Data file.

and car pileups during traffic accidents. In such cases, the design of heterogeneous material with a high energy absorption ratio in one specific direction is meaningful. The P set and their optimized results arrayed with 4 unite cells in *x*-, *y*-, and *z*- directions were tested to examine their mechanical performance in energy absorption. The size for all the models was 60 mm × 60 mm × 60 mm. The tests were carried out using uniaxial dynamic compression at a constant speed of 2 mm/ms, and the time duration was 20ms, leading to a compression of 66.67% of the whole model (40 mm, 2/3 of the *z*-direction length). The constitutive elastoplastic relationship is given in Supplementary Table 9. The prescribed yield strength is 400 MPa. Figure 5a displays the arrayed models and their elastoplastic deformations and collapse modes (also see deformation pattern evolutions from the Supplementary Movie 1–8 in supplementary files). As shown, there are still many local regions under elastic deformation for the original Schwarz P model, making the lattice less efficient in energy absorption. P-4 and -5 as well as their optimized results, for example, are almost yielded all over the whole lattice, ensuring that all materials are fully utilized. In terms of the force-displacement curves during the whole impact period (Fig. 5b), the forces of P-4 and -5 are much larger compared with other models in the P set, indicating that the plastic behavior of a lattice is the key significance for performance in energy absorption. After optimization, the forces of Opt-P-1 and -2 show decreases, but remain stable during the whole impact. The forces of Opt-P-4 and -5

increase further but with observable fluctuations, indicating that clear buckling of components occurred during the impact. As we integrate the force-displacement curves, we can obtain the energy absorption magnitudes. The comparison of normalized energy absorption by P-1 (100%) is presented in Fig. 5c. As can be seen, P-2, Opt-P-1, and -2 show deterioration in energy absorption performance as they suffer more from the bending-dominant mechanical mode in large geometrical deformation. While benefiting from the high stiffness, P-4 and -5 as well as their optimized results are prominent in resisting dynamic loads. Especially for the Opt-P-4 lattice, the energy absorption rate increased by 136% compared with P-1. Thus, it is useful to design and optimize multilayer beam-plate-shell-combined lattice metamaterial considering large geometrical deformation. In this paper, we focus on improving the stiffness of the metamaterial, thereby increasing the resisting force and enhancing the energy absorption performance. However, energy absorption is a dynamic process, and the stability of the model during the impact needs to be taken into consideration. By boosting stability, structural buckling can be alleviated under compression, as a result, the material can be fully utilized and the energy absorption performance can be further improved.

## Isotropic elasticity
Besides improvements in stiffness and strength from the multilayer strategy, homogeneous properties can be achieved from multilayer

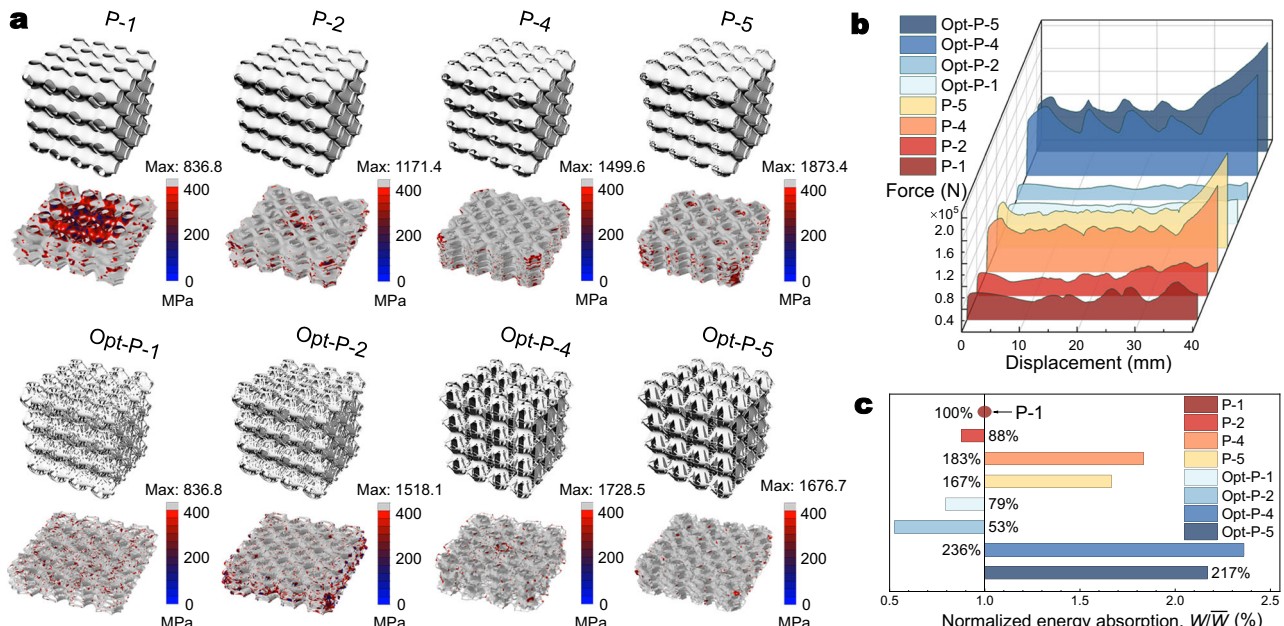

**Fig. 5 | Elastoplastic property of the P set and their optimized results.**
**a** Elastoplastic deformation and collapse mode. The gray regions indicate yielded areas. Maximum Mises stresses are independently provided along the color bars. **b** Force-displacement comparison during compression. **c** Normalized energy absorption rate comparison. Source data are provided as a Source Data file.

configuration. Regarding Young's moduli in three principal directions, the original Schwarz P is stiff in [110] and [111], but weak in [001] directions. While for shear moduli in three principal directions, the original Schwarz P is stiff in [001], but weak in [110] and [111] directions. Thus, to tune the anisotropy property of Schwarz P, one can enhance the modulus in [001] direction, and reduce moduli in [110] and [111] directions. For a cubic symmetry unit cell, the Zener ratio can be used to quantify its anisotropy. When a lattice is homogeneous, its Zener ratio equals 1. From the mathematical expression, the isotropy can be achieved through tuning the Young's and shear moduli and Poisson's ratio. As displayed in Fig. 6, among the P set, P-2 basically has the same Young's and shear moduli and Poisson's ratio with P-1, as a result, they show fewer differences in their Zener ratio. P-4 and P-5 show increases in Young's modulus and decreases in shear modulus as well as Poisson's ratio. Thus, their anisotropy properties are alleviated compared with P-1. P-5's Zener ratio is almost equal to exact 1. Both Young's and shear modulus surfaces are close to a sphere. Note here each layer inside P-5 has the same thickness. By further tuning the thickness of each layer, the anisotropy can be further tuned.

## Tunability

The introduction of multilayer strategy and topology optimization brings a series of tunable dimensions to facilitate the design of multiscale heterogeneous materials. As illustrated in Fig. 7a, a stiffness space can be constructed through a series of tunable dimensions. As such, one can design lattices with desired performance by simply assembling unit cells with specific properties as building blocks. For example, a candidate single-layer model with a tunable area fraction can be designated to construct stiffness-tunable lattices. Figure 7b displays several Schwarz P-based unit cells with different area fractions. Those unit cells were obtained through topology optimization. Figure 7c gives their optimization processes. As the area fraction decreases, the compliance increases. As demonstrated in Fig. 7d, for optimized results, the relationship between stiffness and area fraction shows clear linearity in both situations of the same thickness but different masses and the same mass but different thicknesses. Exploiting this property, one can simply assemble grading microcells with grading area fractions to form functionally graded lattices (Fig. 7e). This can be useful in designing high-

porosity artificial bone (Fig. 7f). Since the stiffness of the bone can be tunable, it stands a better chance to be adaptive to real human body circumstances just like real human bones do. Another important tuning dimension to achieve variable stiffness is the thickness of the lattice. However, the optimized result is thickness-sensitive. Our findings show that the optimized topological configuration becomes more complex when the thickness of the shell decreases. As presented in Fig. 7g and Fig. 7h, optimization with two loading scenarios were studied, i.e., uniaxial, and hydrostatic loadings. One can observe that more intricate components were shaped as the thickness became thinner. This is understandable because more components indicate more load-transferring paths. As the thickness of the shell decreases, the stiffness diminishes. Thus, the optimized results generate and distribute more routes to resist external loads, which is a trade of path number for stiffness. Interestingly, but not surprisingly, we found that the optimized solutions resembled the wing configuration of many insects, such as the dragonfly's wing (Fig. 7i). Due to the ultrathin thickness of the wing, a dragonfly must grow a sophisticated vein structure to support the wing in flight. Here we illustrate how mechanically tunable lattice can be realized with alterable area fraction and thickness. Note that more tunable dimensions can be activated with the introduction of the multilayer strategy and topology optimization.

## Multiphysics extensions

Apart from mechanical design, the multilayer strategy also brings exciting prospects when extending to other physics-based designs of metamaterials. Figure 8 compares the performances of a monolayer and a bilayer lattice (Fig. 8a) in dealing with acoustics (Fig. 8b), electrostatics (Fig. 8c), and flow-thermal coupling problems (Fig. 8d). Material properties are defined in Supplementary Table 10 and Supplementary Table 11. As shown in Fig. 8e, f, the monolayer and bilayer models present different sound pressure distributions under the same sound source. The inner layer works as a wall blocking the transmission of sound. As a result, the total acoustic pressure on the external layer is near zero. In this sense, the multilayer strategy can help create a series of cavities, leading to more freedom for acoustic tuning. This is useful in sound absorption for designing acoustic devices[20]. For the electrostatic problem, the comparison of the monolayer (Fig. 8g) and bilayer model

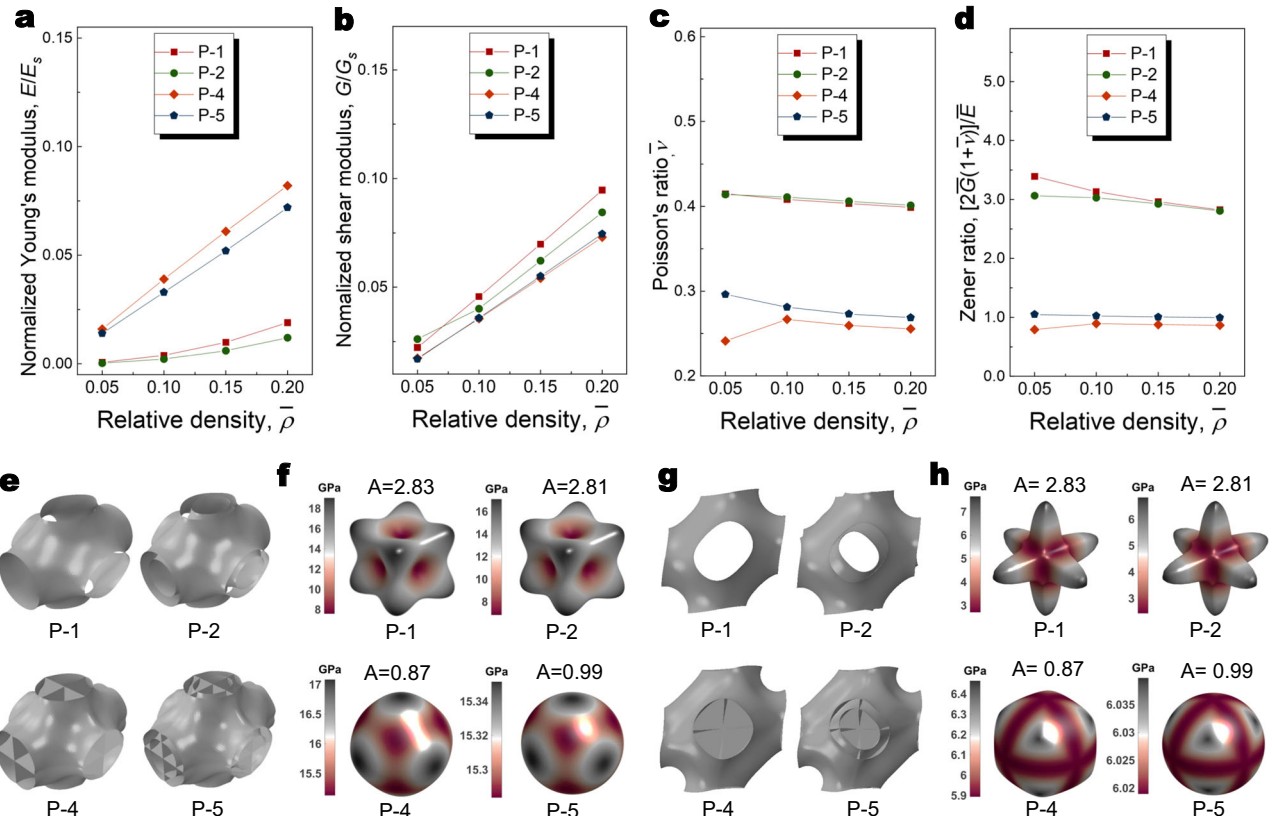

**Fig. 6 | Elasticity property of the P set.** Subfigures (**a**)–(**d**) compare the normalized Young's modulus, normalized shear modulus, Poisson's ratio, and Zener ratio of the P set, respectively. **e** Uniaxial deformation of the P set. **f** Young's modulus of the P set as a function of spatial orientation. **g** Shear deformation of the P set. **h** Shear modulus of the P set as a function of spatial orientation. Source data are provided as a Source Data file.

(Fig. 8h) shows different distributions in electric field norm as well as electric potential. Since the floating potential is applied to model the metallic electrode, one can see that the electrostatic shielding is realized and the electric field norm outside the inner layer is zero. This can be useful in designing electrodes for solid-state batteries[51]. For the flow-thermal coupling problem, the temperature fields for the monolayer (Fig. 8i) and bilayer model (Fig. 8j) are basically the same as the conduction of heat converges into a steady state. However fluid velocity fields for the two models show different distributions, leading to different pressure levels within the lattice. Obviously, the multilayer configuration can change the flow manner. This can lead to a new perspective in fluid field tuning, especially in the design of air-based actuators, such as pneumatic flexible tentacles of soft robots[52]. As discussed above, the multilayer strategy brings new possibilities for metamaterial design considering various physics objectives.

## Discussion

The main contribution of this work lies in providing a method combining a multilayer strategy and topology optimization to design lattice metamaterial. Due to the expanded and diversified design space and freedom, optimized multilayer designs at the same relative density can achieve better mechanical performance. Numerical simulations and physical experiments demonstrated that the obtained beam-plate-shell-combined lattice can achieve ultrahigh stiffness.

In addition to the larger and diverse design space and better results, the multilayer strategy and topology optimization also bring excellent tunability to the design of lattice metamaterial. Available tunable dimensions include shape, thickness, layer number, multilayer configuration, and area fraction, making the design of lattice flexible and diverse. Benefiting from the high tunability, particular mechanical properties can be conveniently realized, such as isotropic elasticity, and

functionally grading stiffness. Also, the multilayer strategy regularizes and tailors the design space. Another advantage is that each layer itself and the cavities separated by those layers can be defined as independent sub-design spaces, which can be flexible in extensions to various problems. For example, apart from mechanical design, the design method in this study shows exciting prospects when extending to the multiphysics-based design of metamaterials.

Despite advantageous properties accompanied by the multilayer strategy and topology optimization, side effects can also occur. For example, topology optimization naturally creates holes on the surface of a lattice, destroying the continuity of the surface so that more stress concentrations may be induced as the loads resisting mode changes. Also, some designs regarding specific problems require airtightness or watertightness, such as the design of pneumatic actuators. In our future work, we explore to design multilayer shell-based lattice whose thickness is variable across the whole surface with minimum limit constraint. As a result, the optimized results can achieve better performance meanwhile maintaining continuous surface. In this way, the multilayer-based design techniques can be further enriched and more choices are available for various design purposes in practical application.

## Methods

### Multilayer construction

The multilayer lattice metamaterial is built upon TPMS, which can be generated through mathematical expressions. For example, the explicit term for the Schwarz P is stated as follows,

$$t = \cos X + \cos Y + \cos Z \quad (1)$$

where $t$ is the mean curvature, $X = 2\pi x/a$, $Y = 2\pi y/a$, $Z = 2\pi z/a$, and $a$ is the unit cell parameter. The mean curvature of TPMS vanishes at

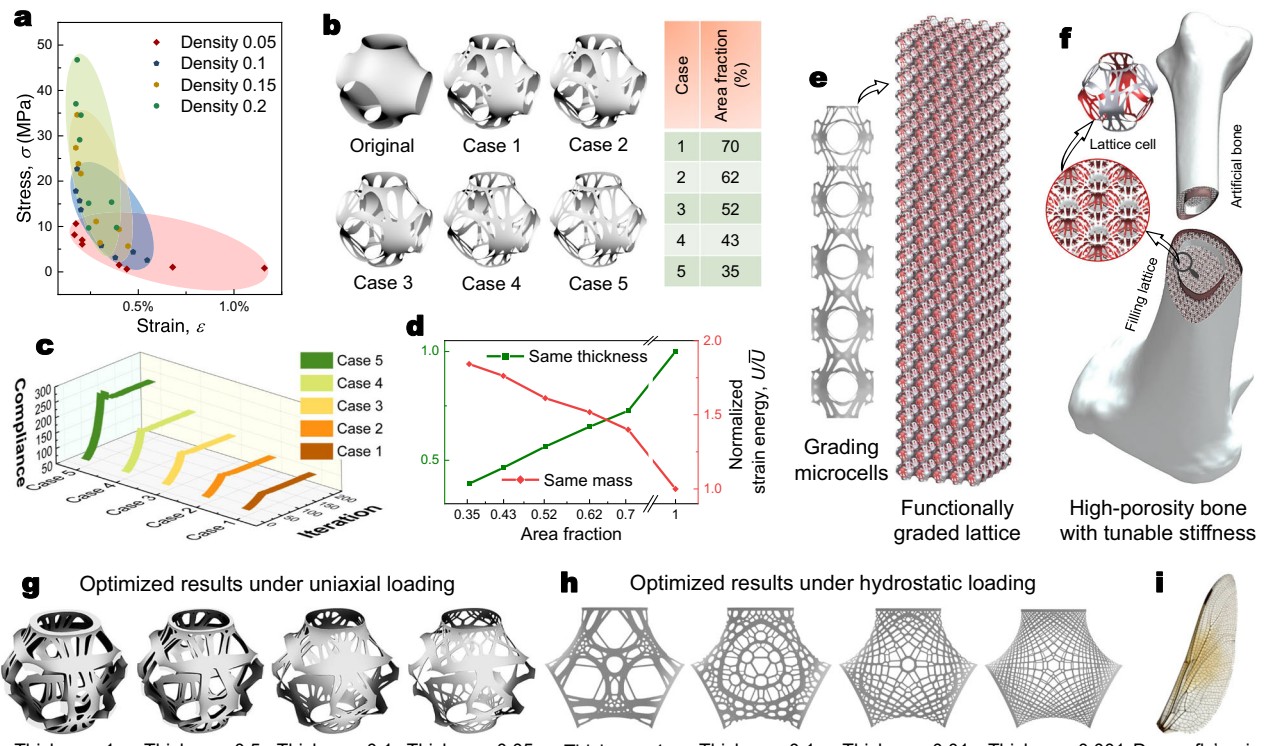

**Fig. 7 | Stiffness-tunable lattice. a** Property space with different relative densities. **b** Area fraction-based tunable dimension for a single layer of the Schwarz P surface. **c** Optimization iteration process of a single layer of the Schwarz P surface with different prescribed area fractions. **d** Relationships between normalized strain energy and area fraction with the same thickness but different masses, and the same mass but different thicknesses. **e** An illustration for a functionally graded lattice with grading micro cells. **f** An illustration of a high porous artificial bone filled with optimized tunable lattices. **g** Optimized results with different thicknesses under uniaxial loading. **h** Optimized results with different thicknesses under hydrostatic loading. **i** A dragonfly's wing. Source data are provided as a Source Data file.

every point on the surface, that is, the Schwarz P is obtained when $t = 0$. The multilayer strategy can be realized through scaling, transforming, and hybridizing the candidate surface. In this way, the design space is expanded and the design freedom is increased. For example, the model P-2 is constructed through a transformation from Schwarz P by changing the value of $t$. By further hybridizing three planes along $x$-, $y$-, and $z$-directions into the P-2, P-5 can be attained.

## Topology optimization

Topology optimization is a powerful technique to redistribute material to achieve optimal design with certain constraints. Other than optimizing the structural configuration, topology optimization can naturally introduce tunable dimensions, such as area fraction, making the design of multilayer shell-based metamaterial more flexible and diverse. In this study, we aim to minimize the compliance of the whole lattice. The optimization mathematical model can be formulated as follows,

$$
\begin{cases}
\min_{\rho_1,\cdots,\rho_{N_e}} : C(\rho_e) = \sum_{e=1}^{N_e} \rho_e^{\,p} \mathbf{u}_e^T \mathbf{K}_e \mathbf{u}_e \\
s.t. \begin{cases}
\mathbf{KU} = \mathbf{F} \\
\sum_{e=1}^{N_e} \rho_e v_e \le A_f \\
t \le T_h \\
0 \le \rho_e \le 1, (e=1,\cdots,N_e)
\end{cases}
\end{cases}
\tag{2}
$$

where $C$ is the objective function of static compliance, $N_e$ the total number of finite elements in the admissible design domain, $\mathbf{u}_e$ the

elemental displacement field, $\mathbf{K}_e$ the elemental stiffness matrix, $\mathbf{K}$ the global stiffness matrix, $\mathbf{U}$ global displacement field, $\mathbf{F}$ the external loads, $A_f$ the allowable area fraction, $t$ the thickness, and $T_h$ the prescribed maximum thickness. The design variable $\rho_e$ is updated with lower and upper bounds of 0 and 1 respectively. The optimization problem can be efficiently solved using an ordinary differential equation (ODE) driven level-set density method[53].

Detailed information on the material characterization, optimization algorithm, practical implementation, as well as method verification for this study, is referred to in the SI text.

## Fabrication and physical experiment

The designed models in this paper were fabricated using additive manufacturing technology. All lattice cell and array models were printed from commercially available 3D printers (SLS PU 500 machine for thermoplastic polyurethane (TPU) and nylon (PA12) materials, BLT S310 machine for stainless steel (ss316) and aluminum alloy (AiSi10Mg) materials). During fabrication, the printing layer thickness was up to 0.1 mm. At least 3 samples were fabricated for each architecture with specific geometric parameters. Actually, to reproduce results in this paper, it does not matter what kind of material is used as we studied the inner proportional relationships among a set of models, and the proportional relationship should be material independent.

The physical experiments were conducted to evaluate the constitutive relationship of printed models. Specifically, uniaxial compression tests were performed along the printing direction of fabricated lattices via a universal testing machine (INSTRON 34FM-100) at a constant nominal strain rate of $5 \times 10^{-3}$ s$^{-1}$

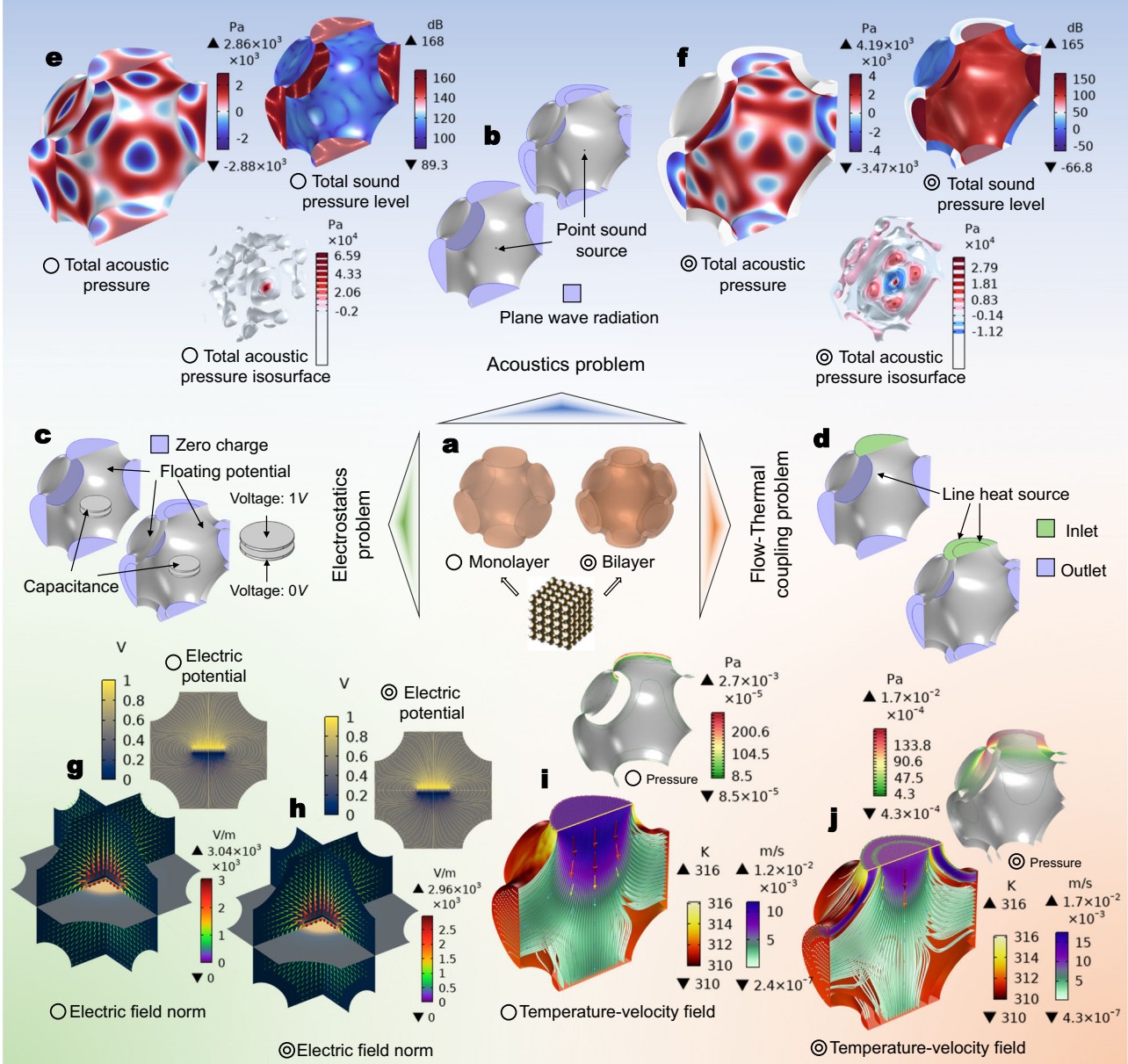

**Fig. 8 | Extension of the multilayer strategy to multiphysics problems.**
**a** Illustration of a single layer and a bilayer cell model. **b**–**d** Model definitions for an acoustic problem, an electrostatic problem, and a flow-thermal coupling problem. **e**, **f** Finite element analysis results of a single layer and a bilayer cell models for the defined acoustic problem, including the total acoustic pressure, total sound pressure level, and total acoustic pressure isosurface. **g**, **h** Finite element analysis results of a single layer and a bilayer cell models for the defined electrostatic problem, including the electric field norm, and electric potential. **i**, **j** Finite element analysis results of a single layer and a bilayer cell models for the defined flow-thermal coupling problem, including the temperature-velocity field, and pressure.

(see Supplementary Fig. 13). The engineering stress can be calculated by dividing the force by the cross-sectional footprint area of the overall sample and the engineering strain was attained by normalizing the displacement by the initial length of the test model. With the obtained constitutive results, Young's moduli can be extracted by fitting the slope of the linear regime of the engineering stress–strain curve.

## Data availability
The data generated in this study are provided in the Source Data file. Source data are provided with this paper.

## Code availability
All necessary information to generate the code used to evaluate the conclusions in this study is present in the paper and Supplementary Information. The original homemade code[54] has been deposited in the Github repository under accession code https://doi.org/10.5281/zenodo.10556426. The reader can download and run it with guidance from the readme document.

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

## Acknowledgements

We gratefully acknowledge the financial support provided by the National Natural Science Foundation of China (Grant Nos. 12272200) and the project of Beijing OptFuture Technology Co., Ltd (No. 20212002316). The first author Yang Liu appreciates very much the valuable discussion and constructive suggestions from Professor Ole Sigmund from the Technical University of Denmark. Also, the first author Yang Liu would like to thank Professor Peng Wei from South China University of Technology for his support and help to the work.

## Author contributions

J.D. designed, guided and supervised the whole research; Y.L. and Y.W. conceived the original idea; Z.L., L.W, W.C., and Y.W. also provided valuable guidance; Y.L., Z.L., Y.W., H.R., X.C. developed the framework of modeling, meshing, and numerical simulation; Y.L., Z.L., H.R., and X.C. performed the optimization and Z.L. contributed a lot to the homemade MATLAB source code; Y.L. and Z.M. designed and conducted the physical experiment; Y.L., Y.W., Z.M., Z.L., and J.D. analyzed the experimental data; Y.L. drafted the initial manuscript with help from co-authors; Y.L., Y.W., H.R., Z.L., L.W., W.C., Y.W., and J.D. edited and revised the manuscript. All authors participated in the discussion and interpretation of the data.

## Competing interests

The authors declare no competing interests.
