## [Peer Review File · Nature Communications]

REVIEWER COMMENTS

Reviewer #1 (Remarks to the Author):

This manuscript provides a timely account of a significant breakthrough in architected metamaterials, unveiling novel mechanical properties characterized by remarkable stiffness and energy absorption capabilities. The robust numerical computations establish a firm groundwork for the proposed concept, while the integration of 3D printing technology further reinforces these findings. The meticulously designed methodology, coupled with the emergence of novel phenomena, underpinned by a compelling argument, has unequivocally convinced me of the urgency of publishing this manuscript in a rapid manner, thereby offering valuable insights to the broader research community. Nonetheless, before final acceptance, I kindly request the authors to address the following raised queries:

(1) In Figure 3, a noticeable disparity between simulations and experimental results is evident. The authors should provide a detailed explanation for this variance, shedding light on the underlying reasons.

(2) The authors have delved into the topic of energy absorption in the proposed design. However, it has been observed that the structures were compressed by less than 10%, which might be insufficient for a comprehensive evaluation of energy absorption. An elaboration on this limitation is necessary.

(3) Figure 5 presents data on energy absorption in the proposed designs. To enhance the comprehensibility of the findings, the authors are encouraged to include a discussion on the mechanisms responsible for energy dissipation, as well as the evolution of deformation patterns. This additional insight will provide a more thorough understanding of the results.

Reviewer #2 (Remarks to the Author):

The authors propose a novel topology optimization method for designing multilayered mechanical structures. Some experimental results suggest that the proposed multilayered designs are superior to non-optimized structures in terms of static and dynamic mechanical properties. I find the manuscript to be well-written and interesting to a wide audience. However, the authors should address the following comments before I recommend the manuscript for publication in Nature Communications:

1. Although I understand that the manuscript focuses on cellular lattice structures, I think that the authors should explain the advantages of the proposed multilayer strategy compared to standard optimized designs. Specifically, why do the authors need the multilayer strategy to increase the design

flexibility? For example, to the best of my knowledge, the standard and conventional SIMP method provides maximum design flexibility.

2. In Figure 2a, the left-most Opt-IWP performs worse than the non-optimized IWP. Please add a discussion on this anomaly.

3. I do not comprehend how to interpret Figure 3b and Figure 4c, 4d. The authors present the experimental results for each model (P set and Opt-P set) with four different printing materials. However, the numerical results are presented for only one case. In this numerical simulation, which printing material is assumed? From these results, I do not observe any consistency between the experimental and numerical results, despite the authors claiming that "... shows much consistency with corresponding numerical results, yet ...".

4. In Supplementary information, please add some references or more detailed descriptions on the material characterization. For instance, how the effective elastic tensor, Zener's ratio, etc. are derived or determined.

5. There are typos in the section title and Eq.(8) in Supplementary information.

Response to the comments of the reviewers

REVIEWER COMMENTS

Reviewer #1 (Remarks to the Author):

This manuscript provides a timely account of a significant breakthrough in architected metamaterials, unveiling novel mechanical properties characterized by remarkable stiffness and energy absorption capabilities. The robust numerical computations establish a firm groundwork for the proposed concept, while the integration of 3D printing technology further reinforces these findings. The meticulously designed methodology, coupled with the emergence of novel phenomena, underpinned by a compelling argument, has unequivocally convinced me of the urgency of publishing this manuscript in a rapid manner, thereby offering valuable insights to the broader research community. Nonetheless, before final acceptance, I kindly request the authors to address the following raised queries:

(1) In Figure 3, a noticeable disparity between simulations and experimental results is evident. The authors should provide a detailed explanation for this variance, shedding light on the underlying reasons.

(2) The authors have delved into the topic of energy absorption in the proposed design. However, it has been observed that the structures were compressed by less than 10%, which might be insufficient for a comprehensive evaluation of energy absorption. An elaboration on this limitation is necessary.

(3) Figure 5 presents data on energy absorption in the proposed designs. To enhance the comprehensibility of the findings, the authors are encouraged to include a discussion on the mechanisms responsible for energy dissipation, as well as the evolution of deformation patterns. This additional insight will provide a more thorough understanding of the results.

Reviewer #2 (Remarks to the Author):

The authors propose a novel topology optimization method for designing multilayered mechanical structures. Some experimental results suggest that the proposed multilayered designs are superior to non-optimized structures in terms of static and dynamic mechanical properties. I find the manuscript to be well-written and interesting to a wide audience. However, the authors should address the following comments before I recommend the manuscript for publication in Nature Communications:

(1) Although I understand that the manuscript focuses on cellular lattice structures, I think that the authors should explain the advantages of the proposed multilayer strategy compared to standard optimized designs. Specifically, why do the authors need the multilayer strategy to increase the design flexibility? For example, to the best of my knowledge, the standard and conventional SIMP method provides maximum design flexibility.

(2) In Figure 2a, the left-most Opt-IWP performs worse than the non-optimized IWP. Please add a discussion on this anomaly.

(3) I do not comprehend how to interpret Figure 3b and Figure 4c, 4d. The authors present the experimental results for each model (P set and Opt-P set) with four different printing materials. However, the numerical results are presented for only one case. In this numerical simulation, which printing material is assumed? From these results, I do not observe any consistency between the experimental and numerical results, despite the authors claiming that "... shows much consistency with corresponding numerical results, yet ...".

- (4) In Supplementary information, please add some references or more detailed descriptions on the material characterization. For instance, how the effective elastic tensor, Zener's ratio, etc. are derived or determined.
- (5) There are typos in the section title and Eq.(8) in Supplementary information.

Detailed response to the Reviewers

The authors are grateful to the editors and reviewers for your time and effort spent on reviewing our manuscript. We would like to thank you very much for your recognition to the manuscript, and your constructive and insightful comments. The comments are very helpful for improving the quality of the manuscript. All comments and questions have been carefully taken into consideration, and we have revised our manuscript accordingly. The amendments on the paper are indicated in red. The specific responses to each point raised by the reviewers are itemized below.

Reviewer #1 (Remarks to the Author):

This manuscript provides a timely account of a significant breakthrough in architected metamaterials, unveiling novel mechanical properties characterized by remarkable stiffness and energy absorption capabilities. The robust numerical computations establish a firm groundwork for the proposed concept, while the integration of 3D printing technology further reinforces these findings. The meticulously designed methodology, coupled with the emergence of novel phenomena, underpinned by a compelling argument, has unequivocally convinced me of the urgency of publishing this manuscript in a rapid manner, thereby offering valuable insights to the broader research community. Nonetheless, before final acceptance, I kindly request the authors to address the following raised queries:

Response: The authors would like to thank the reviewer for the comprehensive and supportive comments which give us great encouragement. We have tried our best to address these issues one by one as follows.

Comment (1): *In Figure 3, a noticeable disparity between simulations and experimental results is evident. The authors should provide a detailed explanation for this variance, shedding light on the underlying reasons.*

Response: Thanks for the suggestion. In the physical experimental part, we aimed to test the mechanical properties of the printed models. We only care about the relative proportional ratios of Young's moduli of the P set and the Opt-P set, which means that the experimental results should be material-independent. Thus, we chose four different printing materials to verify the simulation results. In our numerical simulation, the material iron is used (Young's modulus: 210GPa, yield strength: 400MPa), and the material properties are verified using both static (**Fig. R1a**) and dynamic methods (**Fig. R1b**). Here, different printing materials have different mechanical properties, plastic, brittle, or quasi-brittle. The yield strength can vary for different materials. Therefore, we only consider the effective Young's moduli of the printed models with the different materials, i.e., the stiffness, which can be obtained by calculating the slope of the linear part of the constitutional engineering stress-strain curve (**Fig. R2**). **Table R1** summarizes the effective Young's moduli of the printed models. **Table R2** presents the percentages of effective Young's moduli of the printed models

Fig. R1 | Material property verification. a, static method. b, dynamic method.

Fig. R2 | Linear fit for effective Young's modulus with different materials. a, TPU. b, PA12. c, SS316. d, AISi10Mg.

normalized by the result of P-1. Different printing materials may result in different normalized values, but the **tendency** is consistent with the simulation results. As can be seen in **Table R2**, the proportional ratios among the P set and Opt-P set for the non-metal model (TPU and Nylon (PA12)) show larger values as a whole, but the variation tendency compared with the normalized P-1 is consistent. It seems that the variance between the experiment and the simulation is a proportional factor. For example, if the effective Young's moduli of the TPU result (except P-1) multiply a proportional factor of 75.3%, we obtain results very close to the numerical simulation results (**Table R3**). If the effective Young's moduli of the PA12 result (except P-1) multiply a proportional factor of 52.3%, we obtain results very close to the numerical simulation results (**Table R3**). For the metal result (SS316, AISi10Mg), the proportional ratios show smaller values for P-4, -5 and Opt-P-4, -5. The variances might be caused mainly by printing technics and prototyping quality [1]. For the metal printing technology that requires high temperatures, the residue stress from cooling down from high temperatures may lead to many small defects and cracks, as a result, weakening the mechanical properties of the printed model [1]. For cell models, the local printing quality may affect the experimental results to some extent. For the scale of the 4X4X4 case, the local defects can be alleviated significantly (**Fig. R3**). For the PA12 model, the printed scale is 120mmX120mmX120mm. Such a length scale can ensure better printing quality (**Fig. R4a**). As a result, the experimental results are consistent with corresponding simulation results (**Table R4**). The FBC-Opt-P-2 model is an exception because the mass loss (around 27.4g (17.8%) compared with the P-1) is significant after removing the support

Fig. R3 | Linear fit for effective Young's modulus with different materials. a, PA12 (FBC). b, PA12 (PBC). c, AISi10Mg (FBC). d, AISi10Mg (PBC).

structures and powder remainder. For the AISi10Mg model, since large-size printing for metal models can be very expensive, we printed the metal models with the scale of 60mmX60mmX60mm (**Fig. R4b**), and the printing quality is relatively lower (large-size printing for metal models can be very expensive). Again, the metal results (AISi10Mg) of Opt-P-4 and -5 cannot reach the simulation level due to the reason of printing technics itself and prototyping quality (**Table R4**).

We have revised the 'Results' part in the main text, and the 'Supplementary Note 6' in SI as follows,

- '... uniaxial stiffness in linear elasticity ...'
- '... Thus, the relative proportional ratios of Young's moduli of the P set and Opt-P set should be material-independent. In comparison with the numerical simulations (**Figure 3b**), the basic trend of the physical experimental results shows consistency with corresponding numerical results, yet still some minor differences remain for different kinds of printing material. The discrepancies are mainly caused by printing technics and prototyping quality (detailed discussion in Supplementary Note 6 in SI) ...'
- '... Supplementary Note 6: Disparity between experiment and simulation ...'

Table R1 Numerical and experimental Young's moduli comparison for fabricated cell models (MPa)

Model	Numerical simulation	TPU	PA12	SS316	AlSi10Mg
P-1	813.24	0.064	1.83	320.40	176.00
P-2	465.46	0.039	1.79	258.05	103.83
P-4	8087.7	0.98	34.8	1403.60	993.67
P-5	6842.8	0.69	31.5	1574.31	781.48
Opt-P-1	1880.3	0.21	8.85	634.24	404.19
Opt-P-2	917.09	0.085	3.65	431.42	212.25
Opt-P-4	12607	1.65	59.35	1938.88	1472.81
Opt-P-5	10274	1.09	45.8	2001.96	1350.25

Table R2 Normalized Young's moduli comparison for fabricated cell models

Model	Numerical simulation	TPU	PA12	SS316	AlSi10Mg
P-1	100.0%	100.0%	100.0%	100.0%	100.0%
P-2	57.2%	60.9%	97.8%	80.5%	59.0%
P-4	994.5%	1531.3%	1901.6%	438.1%	564.6%
P-5	841.4%	1078.1%	1721.3%	491.4%	444.0%
Opt-P-1	231.2%	328.1%	483.6%	198.0%	229.7%
Opt-P-2	112.8%	132.8%	199.5%	134.7%	120.6%
Opt-P-4	1550.2%	2578.1%	3243.2%	605.1%	836.8%
Opt-P-5	1263.3%	1703.1%	2502.7%	624.8%	767.2%

Table R3 Normalized Young's moduli comparison for fabricated cell models

Model	Numerical simulation	TPU	PA12
P-1	100.0%	100.0%	100.0%
P-2	57.2%	45.9%	51.1%
P-4	994.5%	1152.9%	994.3%
P-5	841.4%	811.8%	900.0%
Opt-P-1	231.2%	247.1%	252.9%
Opt-P-2	112.8%	100.0%	104.3%
Opt-P-4	1550.2%	1941.2%	1695.7%
Opt-P-5	1263.3%	1282.4%	1308.6%

Table R4 Normalized Young's moduli comparison for fabricated 4X4X4 models

Model	Numerical simulation	PA12-FBC	AlSi10Mg-FBC	Numerical simulation	PA12-PBC	AlSi10Mg-PBC
P-1	100.0%	100.0%	100.0%	100.0%	100.0%	100.0%
Opt-P-1	61.6%	64.0%	95.8%	95.2%	112.5%	94.0%
Opt-P-2	40.7%	17.6%	58.8%	83.6%	65.9%	92.0%
Opt-P-4	383.7%	366.3%	198.7%	385.9%	378.9%	218.4%
Opt-P-5	309.3%	295.8%	200.7%	334.5%	327.7%	186.1%

Fig. R4 | Printing quality comparison for 4X4X4 array models fabricated with PA12 and AISi10Mg. a, printed models with PA12. **b,** printed models with AISi10Mg.

Reference

[1] Al-Ketan, O. and Abu Al-Rub, R.K. (2019), Multifunctional Mechanical Metamaterials Based on Triply Periodic Minimal Surface Lattices. Adv. Eng. Mater., 21: 1900524. <https://doi.org/10.1002/adem.201900524>

Comment (2): The authors have delved into the topic of energy absorption in the proposed design. However, it has been observed that the structures were compressed by less than 10%, which might be insufficient for a comprehensive evaluation of energy absorption. An elaboration on this limitation is necessary.

Response: We are sorry for the confusion. The size for the model was 60mmX60mmX60mm. The compressing speed in our simulation was given at 2mm/ms, and the time duration was 20ms, leading to a compression of 66.67% of the whole sample (40mm, 2/3 of the z-direction length). We think this distance is sufficient as some samples have been compressed into compact states, such as the P-4, -5, and Opt-P-4, -5 (**Fig. R5**, the forces keep increasing).

We have revised the 'Results' part in the main text as follows,

- '... The size for all the models was 60mmX60mmX60mm. The tests were carried out using uniaxial dynamic compression at a constant speed of 2mm/ms, and the time duration was 20ms, leading to a compression of 66.67% of the whole model (40mm, 2/3 of the z-direction length) ...'

Fig. R5 | Force-displacement curves for the P set and Opt-P set.

Comment (3): Figure 5 presents data on energy absorption in the proposed designs. To enhance the comprehensibility of the findings, the authors are encouraged to include a discussion on the mechanisms responsible for energy dissipation, as well as the evolution of deformation patterns. This additional insight will provide a more thorough understanding of the results.

Response: Thanks for the suggestion. Energy absorption is a dynamic process. As can be seen from **Fig. R5**, energy absorption is the integral of external force and displacement. By increasing the stiffness of the cellular metamaterial, thereby increasing the resisting force, the energy absorption performance can be enhanced. Also, the stability of cellular metamaterial can make a difference in the impact of dynamic progress. As shown in **Fig. R5**, the force for P-4, and -5 were relatively stable during the impact. For Opt-P-4 and -5, their stiffnesses show evident improvement, as a result, the maximum forces are at a higher level compared with P-4, and -5. However, their forces during the impact demonstrate clear fluctuations compared with P-4 and -5, indicating severe structural buckling is induced with the compression (**Fig. R5**). In other words, the energy absorption performance can be further improved with enhanced structural stability.

We have added a discussion in the 'Results' part of the main text, and the compression movies for the discussed models are provided in the supplementary files, where the deformation patterns for these models can be observed,

- '... (also see deformation pattern evolutions from the Supplementary Movies 1-8 in SI) ...'
 - '... In this paper, we focus on improving the stiffness of the cellular metamaterial, thereby increasing the resisting force and enhancing the energy absorption performance. However, energy absorption is a dynamic process, and the stability of the cellular model during the impact needs to be taken into consideration. By boosting stability, structural buckling can be alleviated under compression, as a result, the material can be fully utilized and the energy absorption performance can be further improved ...'
-

Reviewer #2 (Remarks to the Author):

The authors propose a novel topology optimization method for designing multilayered mechanical structures. Some experimental results suggest that the proposed multilayered designs are superior to non-optimized structures in terms of static and dynamic mechanical properties. I find the manuscript to be well-written and interesting to a wide audience. However, the authors should address the following comments before I recommend the manuscript for publication in Nature Communications:

Response: The authors appreciate the reviewer's positive feedback and constructive comments. Below, we carefully addressed the comments from the reviewer.

Comment (1): Although I understand that the manuscript focuses on cellular lattice structures, I think that the authors should explain the advantages of the proposed multilayer strategy compared to standard optimized designs. Specifically, why do the authors need the multilayer strategy to increase the design flexibility? For example, to the best of my knowledge, the standard and conventional SIMP method provides maximum design flexibility.

Response: Thanks for the comment. The reviewer is correct that the cubic design domain discretized with standard density element enjoys maximum design freedom. It is certain that the conventional SIMP method can theoretically produce optimized results. However, as the SIMP method is typically performed on the standard cubic density element, it is usually difficult to represent or design surface structures, unless a massive amount of fine elements are used. Apparently, this can lead to computational inefficiency. Unlike the standard SIMP method, the proposed multilayer strategy here regularizes and tailors the design space to better accommodate the surface design while ensuring enough flexibility due to the stacking of multiple layers. It also enables the use of shell elements for plane and surface geometries, and the optimization can be efficient. Another advantage of the multilayer strategy is that each layer itself and the cavities separated by those layers can be set as independent design spaces (**Fig. R6**). In this way, these sub-design spaces

Fig. R6 | Multilayer strategy enables for independent design spaces.

can be assigned the same or different material properties, of the same or different boundary conditions, and the optimization can be performed on those sub-design spaces with the same or different objectives, which can be flexible in dealing with a series of ‘multi-’ problems (**Fig. R6**). In addition, the standard SIMP method does not necessarily guarantee the optimized result is open-cell, which might not be printing-friendly. For the multilayer model, the optimized result can always be open-cell as topology optimization creates holes on the surface of each layer. In summary, the multilayer strategy introduced in this paper can be useful in many perspectives.

We have added a sentence in the ‘Discussion’ part of the revised manuscript,

- ‘... Also, the multilayer strategy regularizes and tailors the design space. Another advantage is that each layer itself and the cavities separated by those layers can be defined as independent sub-design spaces, which can be flexible in extensions to various problems ...’

Comment (2): In Figure 2a, the left-most Opt-IWP performs worse than the non-optimized IWP. Please add a discussion on this anomaly.

Response: Thanks for the comment. In comparison to the original model without optimization, we use the thickness-compensation scheme to ensure they are at the same relative density for fair comparison. The thickness compensation, however, does not necessarily guarantee that the thickened optimized results are better than the original models. Firstly, the optimization of shell structures is thickness-dependent. For example, as can be seen in **Fig. R7a**, the optimized results show quite different topological configurations as the thickness changes for the Neovius model. Thus, the optimized result with compensated thickness can be different from the optimized result with the original thickness. On the other hand, topology optimization reconfigures the material distribution, which may lead to a shift of deformation mode, such as the change from stretching-dominated to bending-dominated. This may weaken the mechanical performance of the original model. For example, as can be seen in **Fig. R7b**, the optimized IWP shows clear bending behavior while the Neovius is still stretching-dominated. As a result, the thickness-compensation works well for the Neovius and its stiffness demonstrates remarkable improvement, while the performance of IWP may suffer deterioration. In addition, topology optimization inevitably creates holes on the surface of the designable region (though we can control the area fraction to limit the area of those holes), as we use shell element to optimize and there is only one layer of the shell element, which can induce stress concentration and local deformation around those holes (**Fig. R7c**), thereby significantly impairing the mechanical performance.

To counter the above-mentioned limitations, one way is to use the multilayer solid element to perform the optimization (**Fig. R7d**). Among the multilayer solid elements, each layer can be assigned as designable or non-designable. In this way, the optimization can be implemented without creating holes by deliberately prescribing a non-designable region, thereby maintaining the smooth surface and avoiding stress concentration. Although this may render close-cell of the cellular material, leading to difficulty in printing postprocess, the setting for non-designable regions can always be optional.

We have added a sentence in the ‘Results’ part of the revised main text, and a discussion in the ‘Supplementary Note 5’ in SI,

- ‘... However, digging holes on the surface with topology optimization does not necessarily guarantee better performance. Several factors, such as the thickness, deformation mode, stress concentration and local deformation, may result in structural deterioration (detailed discussion in Supplementary Note 5 in SI) ...’
- ‘... Supplementary Note 5: Optimization anomaly ...’

Comment (3): I do not comprehend how to interpret Figure 3b and Figure 4c, 4d. The authors present the experimental results for each model (P set and Opt-P set) with four different printing materials. However, the numerical results are presented for only one case. In this numerical simulation, which printing material is assumed? From these results, I do not observe any consistency between the experimental and numerical results, despite the authors claiming that “... shows much consistency with corresponding numerical results, yet ...”.

Fig. R7 | Design and optimization strategy for multilayer shell-based lattice. a, optimized results with different thickness. **b**, deformation modes comparison for the optimized IWP and Neovius. **c**, deformation modes comparison for the Neovius without and with holes. **d**, a multilayer strategy for a single-layer model with independent designable or non-designable regions.

Response: We are sorry for the confusion. We only give one simulation result because we study the relative proportional ratios of the effective Young's moduli among the P set and Opt-P set and they should be material-independent. Thus, we chose four different printing materials to verify the simulation results. In our numerical simulation, the material iron is used (Young's modulus: 210GPa, yield strength: 400MPa). The first reviewer raised the same concern, please see the

detailed response to comment (1) from Reviewer 1.

Comment (4): In Supplementary information, please add some references or more detailed descriptions on the material characterization. For instance, how the effective elastic tensor, Zener's ratio, etc. are derived or determined.

Response: Thanks for the suggestion. We have added related discussion in 'Supplementary Note 1', and references in the 'Supplementary References' in SI,

- '... Supplementary Note 1: Material characterization ...'
 - '... Supplementary References ...'
-

Comment (5): There are typos in the section title and Eq.(8) in Supplementary information.

Response: Thanks for the correction. We have revised those typos in the manuscript.

REVIEWERS' COMMENTS

Reviewer #1 (Remarks to the Author):

Accept

Reviewer #2 (Remarks to the Author):

I am satisfied with the authors' response to my review comments. I recommend the revised manuscript for the publication in Nature Communications.

Response to the comments of the reviewers

REVIEWER COMMENTS

Reviewer #1 (Remarks to the Author):

Accept

Reviewer #2 (Remarks to the Author):

I am satisfied with the authors' response to my review comments. I recommend the revised manuscript for the publication in Nature Communications.

Detailed response to the Reviewers

The authors appreciate very much the editors and reviewers' time and effort spent on reviewing our manuscript. Thanks to your constructive and insightful comments, our manuscript has improved a lot.

Reviewer #1 (Remarks to the Author):

Accept

Response: The authors would like to thank the reviewer for the comment.

Reviewer #2 (Remarks to the Author):

I am satisfied with the authors' response to my review comments. I recommend the revised manuscript for the publication in Nature Communications.

Response: Many thanks to the reviewer's recommendation.